# First Record of *Colletotrichum anthrisci* Causing Anthracnose on Avocado Fruits in Chile

**DOI:** 10.3390/pathogens11101204

**Published:** 2022-10-19

**Authors:** Marcelo I. Bustamante, Claudio Osorio-Navarro, Ysadora Fernández, Tyler B. Bourret, Alan Zamorano, José Luis Henríquez-Sáez

**Affiliations:** 1Departament of Plant Health, Faculty of Agricultural Sciences, University of Chile, Avenida Santa Rosa 11315, La Pintana 8820808, Santiago, Chile; 2Department of Plant Pathology, University of California, Davis, CA 95616, USA; 3Plant Molecular Biology Centre, Department of Biology, Faculty of Sciences, Universidad de Chile, Las Palmeras 3425, Ñuñoa 7800003, Santiago, Chile

**Keywords:** etiology, postharvest disease, *Persea americana*

## Abstract

Anthracnose caused by *Colletotrichum* species is one of the most frequent and damaging fungal diseases affecting avocado fruits (*Persea americana* Mill.) worldwide. In Chile, the disease incidence has increased over the last decades due to the establishment of commercial groves in more humid areas. Since 2018, unusual symptoms of anthracnose have been observed on Hass avocado fruits, with lesions developing a white to gray sporulation. Morphological features and multi-locus phylogenetic analyses using six DNA barcodes (*act*, *chs-1*, *gapdh*, *his3*, ITS, and *tub2*) allowed the identification of the causal agent as *Colletotrichum anthrisci*, a member of the dematium species complex. Pathogenicity was confirmed by inoculating healthy Hass avocado fruits with representative isolates, reproducing the same symptoms initially observed, and successfully reisolating the same isolates from the margin of the necrotic pulp. Previously, several *Colletotrichum* species belonging to other species complexes have been associated with avocado anthracnose in other countries. To our knowledge, this is the first record of *C. anthrisci* and of a species of the dematium species complex causing anthracnose on avocado fruits in Chile and worldwide.

## 1. Introduction

Avocado (*Persea americana* Mill.) is a tropical and subtropical fruit crop that originated in Mexico and Central America and has been cultivated and consumed for more than 8000 years [1,2]. In recent decades, avocado fruits have gained interest due to their nutritional composition and health benefits, leading to increased production in subtropical and Mediterranean climate countries [3,4]. In Chile, avocado is an important fruit crop, comprising approximately 32,000 hectares clustered between the Coquimbo and O’Higgins regions [5]. By 2021, the country produced 229,137 tons of avocado fruits, of which 41.5% were exported to Europe (72.4%), Argentina (14.0%), China (7.9%), the USA (4.7%), and South Korea (1.0%) [6]. The most planted and the only exported cultivar is Hass. Even though the crop has been considered susceptible to relatively few diseases [7], postharvest diseases such as anthracnose and stem end rot have been alerting growers and export companies due to the losses reported during the last decade [8].

Anthracnose is a common disease that affects avocados worldwide and has been a model for the study of fungal latent infection on fruits, with infections usually taking place in the orchard and symptoms showing up after fruit ripening [9]. In Chile, the incidence of the disease has increased during the last two decades due to the development of new high-density plantations in areas with high relative humidity, where free water from dew can persist for several hours on the surface of plant tissues. Symptoms of the disease include black and circular lesions over the fruit skin that rapidly grow in diameter and become sunken. On larger and older lesions, orange to pink spore masses emerge from acervuli produced by the pathogen, conferring a waxy aspect to the sporulation [10]. Internally, the pulp turns dark, with the necrotic tissue remaining attached to the fruit skin and being easily removed when peeling the fruit, leaving a characteristic cavity. Consequently, the shelf life and marketability of the fruit is significantly reduced, increasing economic losses for growers and stakeholders.

The causal agents of avocado anthracnose are fungi of the genus *Colletotrichum* [11], and approximately 20 species have been reported worldwide, comprising members of five different species complexes, namely gloeosporioides, acutatum, boninense, gigasporum, and magnum [11,12,13,14,15,16,17]. In Chile, however, only *C. gloeosporioides sensu lato* has been reported [18], which produces orange-colored sporulating lesions. Morales et al. [18] based their identification solely on morphology, 33 years before the genus was taxonomically reorganized and *C. gloeosporioides* was redefined as a species complex [11]. Since then, no other species have been associated with the disease in the country. During the 2018 and 2021 growing seasons, Hass avocado fruits that were harvested from two commercial groves in Central Chile developed unusual symptoms of anthracnose, characterized by a white to grey sporulation on the fruit lesions. The aim of this study was to investigate the etiology of this symptomatology.

## 2. Results

### 2.1. Symptomatology

Black circular lesions developed on the fruit skin after ripening, with one to several lesions per fruit. Over time, lesions increased in size, reaching a diameter between 20 and 30 mm. From the center of lesions, spore masses with waxy aspect developed, exhibiting unusual colors ranging from white to grey (Figure 1C). Approximately 1–5% of the symptomatic fruit developed this type of sporulation and were separated for further inspection. The pulp showed dark brown discoloration underneath the lesions, with a rounded shape toward the center of the fruit. Necrotic pulp separated easily from the healthy tissue, leaving a characteristic cavity. Single-spore isolations resulted in 29 isolates from different fruits that showed consistent morphological features, and three representative isolates were selected for further analyses.

### 2.2. Morphological Analysis

On potato dextrose agar (PDA), fungal colonies developed aerial mycelium with dark gray colors and a white margin. On the reverse, colonies were dark grey to black with bluish tones (Figure 1A). Acervuli were not formed, therefore other media were used. Spezieller Nährstoffarmer agar (SNA) allowed for the slow formation of acervuli in approximately 30 days, however no conidia were observed. A less common medium, casitone-yeast extract agar (CYE) successfully induced acervular development in 7 days at 25 °C (Figure 1B). Mycelium grew flat and hyaline, with vegetative hyphae of 1–8 μm width, smooth walled, septate, and branched. Chlamydospores were not observed. Acervuli were black and formed a ring between the center and the margin of the colonies. Conidia developed white to pale grey masses that oozed from the apex of acervuli. Conidiophores and setae formed on a basal cushion of rounded brown cells from acervuli (Figure 1D). Conidia were aseptate and smooth-walled with curved shape, bent abruptly, tapering towards both ends, hyaline central part frequently nearly straight (Figure 1E)**,** with 2–4 guttules, 21.4 ± 1.4 × 4.7 ± 0.6 μm in size and a L/W ratio of 4.5. Appressoria (Figure 1F) were solitary, in chains or in loose groups, pale brown to brown, aseptate, smooth-walled, navicular to ovoid and occasionally clavate or bullet-shaped, size 10.3 ± 2.8 × 5.6 ± 0.8 μm (mean ± SD), and a L/W ratio of 1.8. Conidiophores were mostly simple but occasionally branched, hyaline to pale brown and septate (Figure 1G). Conidiogenous cells were enteroblastic, hyaline, smooth-walled, cylindrical to ampulliform, 5–22 × 3–5 μm in size, and no collarette was observed. Setae (Figure 1H) were brown to dark brown, two to five septate, 62.5–320 × 4–7.5 μm, base constricted, tip acute, and smooth. Sexual morph was not observed.

On inoculated Hass avocado fruits, acervuli were black, with conidiophores and setae formed from a cushion of brown, angular cells. Conidiophores were hyaline to pale brown, septate, and branched. Conidiogenous cells were enteroblastic, pale brown, cylindrical, 5–20 × 3–5 μm, with an opening of 1.5–2 μm diameter, and collarette was not observed. Conidia were hyaline, smooth-walled, and aseptate, with the central part usually straight with parallel walls, bent abruptly to a strongly acute apex and truncate base, 23–29 × 3–5 μm, mean ± SD = 25.8 ± 1.39 × 4.3 ± 0.59 μm, with a L/W ratio of 6. Setae were dark brown, opaque, two to five septate, 112.5 to 480.0 µm long, with a constricted base, measuring 7.5 to 14.5 µm at the widest part and ending in an acute tip, smooth, with a L/W ratio of 26.9 (Table 1).

### 2.3. Phylogenetic Analysis

The length of the consensus sequences ranged from 252 to 254 bp for *act*, 288 to 301 bp for *chs-1*, 303 to 305 bp for *gapdh*, 387 to 392 bp for *his3*, 494 to 549 bp for ITS, and 631 to 750 bp for *tub2*. Due to the lack of availability of *his3* sequences on closely related species, the maximum likelihood inference included a data set of five loci (*act*, *chs-1*, *gapdh*, ITS, and *tub2*). The resulting tree revealed that Chilean isolates clustered with strains of *C. anthrisci* (Figure 2). Moreover, strains of *C. sambucicola* formed a subclade within this cluster, acting as a sister taxon. In the single-gene phylogenies, the Chilean isolates consistently formed high supported clades with *C. anthrisci* strains (Appendix A), except in the *gapdh* tree, where they clustered with *C. sambucicola* (Appendix A). Additionally, a four-locus analysis (*act*, *gapdh*, ITS, and *tub2*) was performed (Appendix A), which showed the same topology as Figure 2.

### 2.4. Effect of Temperature on Mycelial Growth

The isolates grew under temperatures that ranged from 4 to 30 °C for 11 days, with a maximum growth rate at 20 °C, decreasing thereafter (Figure 3). A similar pattern of growth rate was observed among the isolates; however, the interaction between the isolates and the temperatures was significative (*p* < 0.05), and differences were detected among the isolates at the different temperatures, except at 35 and 40 °C, where no growth was observed. Additionally, when exposed to high temperatures, the mycelium was only able to resume growth after being incubated at 30 and 35 °C, but not at 40 °C. The second-order polynomial Equation (1) yielded minimum, optimum and maximum temperatures of 3.5, 18.8, and 34.1 °C, respectively.
y = −0.0274x^2^ + 0.9929x − 2.3746; R^2^ = 0.8347(1)

### 2.5. Pathogenicity Test

Inoculated avocado fruits developed identical symptoms of anthracnose after 7 days, while the controls remained symptomless (Figure 4). Lesions diameters ranged from 22.3 to 26.7 mm (mean 25.9 mm) for wounded fruits after 8 days, and from 33.0 to 38.6 (mean 35.7) for nonwounded fruits after 20 days. The same fungal isolates were reisolated from necrotic pulp of symptomatic fruit, which was confirmed by morphology. Lesion diameters were not significantly different between isolates. On nonwounded leaves, isolate NAL53 was pathogenic on 66.6% of inoculated leaves, while controls showed no symptoms, after 20 days. Necrotic lesions ranged from 0.2 to 11.2 mm (mean 3.2 mm) and the same fungus was recovered after isolations of the margin of symptomatic tissue.

## 3. Discussion

Over 20 species of *Colletotrichum* have been reported as causal agents of avocado anthracnose worldwide. Different studies suggest a predominance of species belonging to the gloeosporioides species complex, followed by members of the acutatum and boninense complexes [13,14,15,17]. Similarly, *C. gigasporum* and an undescribed species of the magnum complex represent unique detections in Sri Lanka and Mexico, respectively [12,15]. In this study, we report for the first time the occurrence of a member of the dematium complex causing the disease in Chile. Morphologically, this group includes species with curved conidia, spherical to oval appressoria, and dark and stiff setae [19]. Six species were initially described in 2012, indicating a preference toward herbaceous plants [11,19]. However, more species have been discovered during the last decade, including pathogens of woody hosts [20,21,22,23,24,25,26,27]. Currently, 18 species are recognized in this complex, acting as saprobes, endophytes, and pathogens in a range of plant hosts [28]. In this article, we have proven that *C. anthrisci* is responsible for anthracnose on avocado fruits, of which there was no previous evidence. This fungal species has only been detected causing stem lesions on *Anthriscus sylvestris* in the Netherlands [19], and damping-off on seedlings of *Cornus controversa*, *Fraxinus lanuginosa*, *Magnolia obovate*, and *Prunus grayana* in a Japanese forest [29]. Therefore, this is the first record of *C. anthrisci* in the Americas and in the Southern Hemisphere, which indicates that the pathogen has a wider distribution.

The accuracy of fungal species identification is critical for early detection of pathogens, disease prevention, and management [30]. Some species of *Colletotrichum* are considered quarantine pests and therefore, an incorrect identification can lead to ineffective disease control strategies. Other species can also cause human infections [31,32,33,34]. Traditionally, *Colletotrichum* species have been identified using morphological features, host association and phylogenetic analyses, with the latter gaining significant importance in recent years [35]. Here, both morphological and phylogenetic approaches allowed the identification of *C. anthrisci*. Morphologically, the presence of navicular to bullet-shaped appressoria, the size and shape of conidia, and the length and aspect of setae resemble the descriptions of *C. anthrisci* [19]. The five-locus phylogeny showed a highly supported clade between the Chilean isolates and strains of *C. anthrisci* (Figure 2). Interestingly, when revising the single gene phylogenies, the Chilean isolates clustered with *C. anthrisci* strains in the *act*, ITS, and *tub2* trees (Appendix A), except in the *gapdh* tree (Appendix A), where they grouped with *C. sambucicola* (89% bootstrap support value). This is not surprising, since previous studies have found similar incongruences with *C. orchidis* and *C. quinquefoliae*, both members of the dematium complex, where one locus showed a higher identity match with species of other species complexes, resulting in unexpected topologies in the phylogenetic trees [28].

The growth of the Chilean isolates of *C. anthrisci* are favored by medium temperatures with an optimum of 20 °C, which are normal conditions for temperate climates such as those in Central Chile. Similar findings have been described for other members of the dematium complex, suggesting that these species are adapted to temperate climates [11]. Conversely, other species of *Colletotrichum* affecting tropical and subtropical crops with higher optimal temperatures (25–30 °C) have shown to be unable to cause disease in colder areas [36,37,38]. Therefore, the optimal temperature of development could explain the geographical distribution of populations of *Colletotrichum*. Interestingly, we determined that *C. anthrisci* was viable after being previously incubated at 30 and 35 °C, but not at 40 °C, for 11 days. This may suggest that the pathogen can survive and be found in warmer areas, but more studies are needed to determine the geographical distribution of this species.

Finally, pathogenicity was confirmed on wounded fruits and on nonwounded fruits and leaves, with lesions developing slower in the latter. These findings are relevant for management strategies. Symptomatic fruits and leaves serve as inoculum source and can remain unnoticed under the canopy leading to future infections under favorable conditions. The disease significantly reduces the fruit shelf life and marketability, since avocados are mainly sold unripe and the symptoms normally develop after ripening. Therefore, fruit with asymptomatic infections is marketed to consumers who later discover anthracnose, potentially biasing them against future avocado purchases. This is especially true for the export industry, where Chilean avocados are transported by sea in cooling containers (4–8 °C) with controlled atmosphere for 25–45 days and later incubated at room temperature to induce ripening before reaching the supermarket shelf. In China, *C. fructicola* was detected on Hass avocados which were imported from Peru and developed the disease in the supermarket, which raised concerns about quarantine measures [39]. A more drastic situation occurred in Chile, where the quarantine pathogen *Monilinia fructicola* was detected on nectarines imported from California in 2009, and later established in the country after years of eradication efforts [40]. Further investigation is needed to better understand the epidemiological roles of *C. anthrisci* in avocado anthracnose, a disease known to be caused by multiple species of *Colletotrichum*.

## 4. Materials and Methods

### 4.1. Fruit Sampling

During both 2018 and 2021 growing seasons, approximately 1260 healthy-looking Hass avocado fruits were collected from two commercial groves located in Naltahua, Metropolitan Region, and in Llayllay, Valparaiso Region, Chile, respectively. Fruits were randomly collected from bins, trees, and grove floor. In the laboratory, fruits were incubated inside cardboard packaging at room temperature (20 °C) under a light regime of 12 h of fluorescent light and 12 h of darkness and kept at 40–50% humidity for 14 days until anthracnose symptoms were observed.

### 4.2. Fungal Isolation

Single-spore isolations were performed by pipetting 10 μL of sterile distilled water (SDW) onto an individual sporulating lesion and dispensing the resulting suspension in 200 μL of SDW. An aliquot of 100 µL of the conidial suspension was poured on the surface of a water agar plate and spread with a sterile glass spatula. After an incubation of 18 h at 20 °C, a germinated conidium was transferred with a sterile needle onto full-strength potato dextrose agar (PDA) amended with streptomycin (100 mg/L). Each isolate was obtained from different symptomatic fruits.

### 4.3. Morphological Analysis

Three representative isolates (NAL52, NAL53, and NAL54) were cultured on PDA, SNA [41] and CYE [42] plates for seven to thirty days at 20 °C in darkness to induce the formation of different fungal structures. Additionally, ripened Hass fruits were inoculated to evaluate the same structures on the fruit skin. Fruits (*n* = 3) were disinfected with 0.5% sodium hypochlorite for 3 min and rinsed with SDW. Once dried out, fruits were pinpricked at the center of the equatorial diameter with a sterile needle of 0.5 mm diameter at 1 mm of depth and inoculated with 20 µL of a conidial suspension of each isolate. Conidial suspensions were obtained by adding 20 µL of SDW onto a sporulating acervulus from cultures grown on CYE agar, pipetting the resulting solution in 300 µL of SDW, and adjusting the concentration to 10^6^ conidia/mL with a Neubauer chamber. Fruits were incubated in humid chambers (>80% RH) made of plastic trays with moistened paper towels at 20 °C for a period between 7 and 21 days until symptoms development. The morphological structures of the three isolates grown in plates and on fruits were measured in micrometers with a phase-contrast microscope (Carl Zeiss Axiostar Plus, Göttingen, Germany). Conidia, conidiogenous cells, conidiophores, setae, and hyphae were mounted in water, whereas appressoria were observed on the undersurface of the CYE cultures. Thirty measurements of different versions of each structure were carried out, and photographs were obtained with a digital camera (Canon Powershot A640).

### 4.4. DNA Extraction, Amplification of DNA Barcodes, and Sequencing

Selected isolates were cultured on PDA for seven days and mycelium was further frozen and lysed, following the protocol of the Fungi/Yeast Genomic DNA isolation kit (NORGEN; #27300). The internal transcribed spacer of rDNA (ITS) and partial gene regions of actin (*act*), chitin synthase (*chs-1*), glyceraldehyde-3-phosphate dehydrogenase (*gapdh*), histone H3 (*his3*), and β-tubulin (*tub2*) were amplified by PCR, using the primer pairs ITS1/ITS4 [43], ACT-512F/ACT-783R [44], CHS-79F/CHS-354R [44], GDF1/GDR1 [45], CYLH3F/CYLH3R [46], and T1/Bt2b [47,48], respectively. Amplifications comprised a total volume of 25 μL, containing a final concentration of 1X GoTaq^®^ Master Mix (Promega; #M7133), 0.3 µM of each primer, and 50–100 ng of genomic DNA, completing the volume with Nanopure water. The PCR conditions consisted of a 5 min denaturation step at 94 °C, followed by 30 cycles of 30 s at 94 °C, 45 s at 48 °C for ITS, or at 61 °C for *act*, or 58 °C for *chs-1* and *his3,* or 52 °C for *gapdh* and *tub2*, and 40 s at 72 °C, with a final extension of 7 min at 72 °C. PCR products were sequenced by Psomagen USA (Rockville, MD, USA) and consensus sequences were obtained by assembling forward and reverse sequences using the software CAP3 [49].

### 4.5. Phylogenetic Analysis

A multi-locus data set consisting of sequences of six loci (*act*, *chs-1*, *gapdh*, *his3*, ITS, and *tub2*) of *Colletotrichum* species was retrieved from the NCBI database and assembled along with the corresponding sequences of the Chilean isolates, with *C. gloeosporioides* (type strain CBS 112999) and *C. karsti* (type strain CBS 127597) as outgroups (Appendix A). Sequences were separately aligned by locus using MAFFT 7 and selecting the G-INS-i refinement method [50]. Each locus of the concatenated data set was partitioned by coding and non-coding regions (but not by codon position), resulting in 20 partitioned subsets. IQTREE 2.1.3 [51] was used with the following options for maximum likelihood inference: linked branch lengths for partitioned analysis, ‘greedy’ algorithm from PartitionFinder2 [52] for merging partitions, ’merge-model all’ and ‘merge-rate all’ to employ the widest range of evolutionary models, corrected Akaike information criterion for model testing, ‘allnni’ for a more thorough tree search, and 1000 non-parametric bootstrap replicates for support values. IQTREE merged the 20 subsets into 14 subsets. Bayesian inference was conducted using MrBayes 3.2.7 [53], following the methodology of Bourret et al. [54]. Resulting trees were examined and support values were combined with TreeGraph2 [55], with visual edits using Inkscape 0.92 (http://inkscape.org, accessed on 1 September 2022). Trees were also inferred from each locus individually (Appendix A) using the ultrafast bootstrap approximation rather than the non-parametric bootstrapping used for the multi-locus tree.

### 4.6. Effect of Temperature on Mycelial Growth

The three isolates were cultivated in PDA for 7 days at 20 °C, and 3 mm diameter agar plugs with actively growing mycelium were obtained from the edge of the colonies using a sterile cork borer. The plugs were placed on the center of Petri dishes containing full-strength PDA with the mycelium facing the surface of the medium. The plates were incubated under seven temperatures (4, 10, 20, 28, 30, 35 and 40 °C) in darkness for 11 days and the diameter of the colonies were measured daily using a caliper. Two perpendicular measurements were averaged for each colony. The experiment was performed twice, with three replicated plates per isolate, and the data was averaged before growth rate calculations (mm/day). A two-way ANOVA was performed using generalized linear models with the corresponding R packages in InfoStat v2008. Normality and homoscedasticity were checked and corrected when necessary, and means were separated using Fisher’s least significant difference test (*p* < 0.05). Data was plotted in GraphPad Prism v.5.03 for Windows. Additionally, the cultures incubated at 30, 35 and 40 °C were incubated for 7 more days at room temperature (20 °C) to determine the viability of mycelium exposed to high temperatures. The minimum, optimum and maximum growth temperatures for the isolates were estimated using a second-order polynomial equation obtained with scatter plots constructed with the data in Microsoft Excel 365.

### 4.7. Pathogenicity Tests

Each isolate was inoculated in sets of five apparently healthy Hass avocado fruits. Acervuli formation was induced by culturing the isolates on CYE agar for 14 days at 20 °C and conidia were harvested by pipetting 10 µL of SDW on a conidial mass and transferring the resulting solution into 300 µL of SDW. The suspension was adjusted to 10^6^ conidia/mL using a Neubauer chamber. Fruits were disinfected with 0.5% sodium hypochlorite for 3 min and rinsed with SDW. Once dried out, one group of fruits was wounded at the center of the equatorial diameter using a sterile needle of 0.5 mm diameter at 1 mm of depth whereas the second group was not wounded. Inoculations were carried out by adding 20 µL of the conidial suspension on the wound or at the center of the equatorial diameter of each fruit. Control fruits received SDW. Fruits were incubated in humid chambers (>80% RH) made of plastic trays with moistened paper towels at 20 °C for a period between 7 and 21 days until symptom development. Evaluation was carried out by removing the fruit skin and measuring the diameter of necrotic pulp using a caliper. Isolations were carried out by plating tissue pieces of the margin of the lesions on PDA for seven days at 20 °C. Since there were no differences in lesion diameter between isolates, one isolate (NAL53) was selected to inoculate 12 nonwounded leaves following the same methodology used for fruits. Briefly, leaves were disinfected, inoculated, and incubated for 20 days in humid chambers. Lesions were measured at two perpendicular directions and isolations of were performed on PDA from the margin of the necrotic area. Data was plotted in GraphPad Prism v.5.03 for Windows.

## 5. Conclusions

This study represents the first record of *C. anthrisci*, a member of the dematium species complex, causing anthracnose on avocado fruits. The pathogen caused unusual sporulating lesions with white to grey oozing conidial masses, as opposed to the typical orange to pink colors produced by other *Colletotrichum* species associated with the disease. Selected isolates were pathogenic on nonwounded fruits, leaves and wounded fruits. Optimum temperature for mycelial growth was 20 °C, which could explain the distribution in Chilean groves located in areas with mild temperatures (17–30 °C). Further studies are required to better understand the epidemiology of *C. anthrisci* and to develop potential management strategies.

## Figures and Tables

**Figure 1 pathogens-11-01204-f001:**
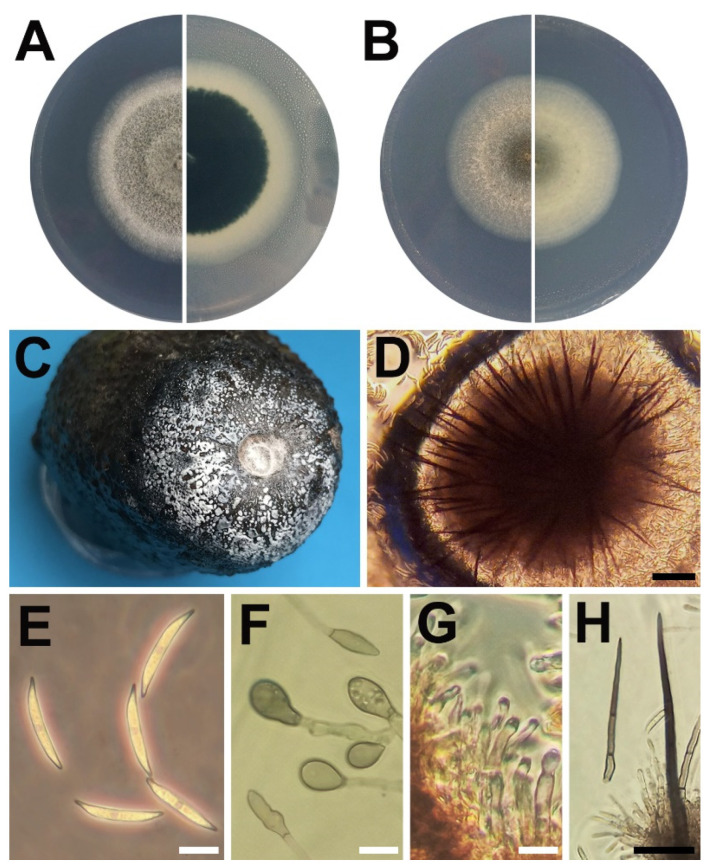
(**A**) Colony aspect of isolate NAL52 growing on PDA after seven days at 20 °C, top view (left) and bottom view (right); (**B**) NAL52 on CYE agar; (**C**) sporulating lesion of anthracnose on Hass avocado fruit; (**D**) acervulus with setae; (**E**) conidia; (**F**) appressoria; (**G**) conidiophores and conidiogenous cells; and (**H**) setae. White bar = 10 µm; black bar = 100 µm.

**Figure 2 pathogens-11-01204-f002:**
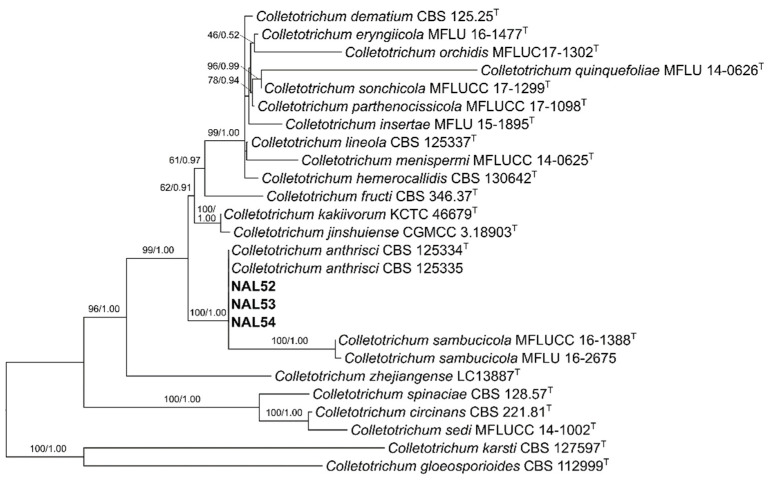
Maximum likelihood phylogenetic analysis of *Colletotrichum* species of the dematium species complex. Chilean isolates are shown in bold. Maximum likelihood bootstrap values and Bayesian posterior probabilities are indicated above the branches. The tree was inferred from a data set consisting of sequences of five loci (*act*, *chs-1*, *gapdh*, ITS, and *tub2*), and rooted to *C. gloeosporioides* (CBS 112999) and *C. karsti* (CBS 127597). Ex-type strains are noted with ^T^.

**Figure 3 pathogens-11-01204-f003:**
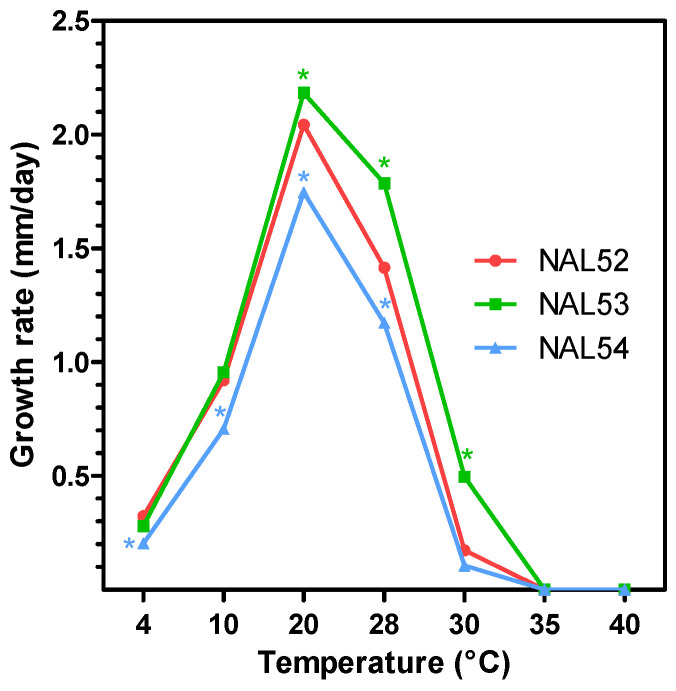
Effect of temperature on the mycelial growth of three isolates of *C. anthrisci* causing avocado anthracnose in Chile. Asterisks represent significant differences between isolates compared to NAL52 at each temperature, according to Fisher’s LSD test (*p* < 0.05).

**Figure 4 pathogens-11-01204-f004:**
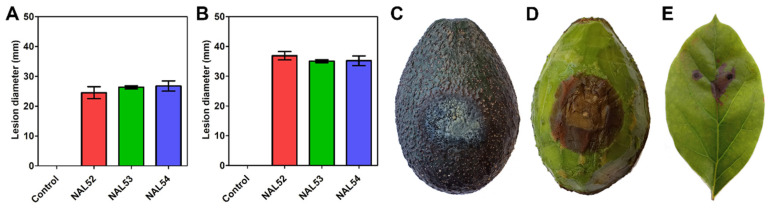
Pathogenicity tests on Hass avocado fruits inoculated with conidial suspensions (10^6^ conidia/mL) of three isolates of *C. anthrisci*. Lesion diameters on wounded fruits after 8 days (**A**) and on nonwounded fruits after 20 days (**B**). Bars show mean and standard deviation of diameters of obtained lesions. External (**C**) and internal symptoms (**D**) of nonwounded fruit. (**E**) Lesions on nonwounded leaf after 20 days.

**Table 1 pathogens-11-01204-t001:** Morphological features (range or average ± standard deviation) of Chilean isolates (NAL), *C. anthrisci*, and *C. sambucicola*.

Character	NAL Isolates ^1^	*C. anthrisci* ^2^	*C. sambucicola* ^3^
Conidia length (μm)	25.8 ± 1.4	25 ± 1.5	17.7
Conidia width (μm)	4.3 ± 0.6	3.5 ± 0.2	3.8
Conidia L/W ratio	6	7.3	4.7
Setae septum (*n*)	2–5	2–4	1–6
Setae length (μm)	112.5–480	90–350	110–145
Setae width (μm)	7.5–14.5	6–18	6.1–8.5
Setae L/W ratio	26.9	18.3	17.5

^1^ Measurements performed on host tissue (fruits of *Persea americana*). ^2^ Data from Damm et al. [19], on host tissue (stems of *Anthriscus sylvestris*). ^3^ Data from Tibpromma et al. [20], on host tissue (dead branches of *Sambucus ebulus*).

## Data Availability

Not applicable.

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
