# Peer review of "First Record of Colletotrichum anthrisci Causing Anthracnose on Avocado Fruits in Chile"

_pathogens, 2022, doi:10.3390/pathogens11101204_

Round 1

Reviewer 1 Report

The manuscript reports on the identification of Colletotrichum anthrisci causing anthracnose of avocado in Chile. This paper includes morphological description of the pathogen, six-gene phylogenetic analysis and pathogenicity bioassays.  The bioassays were based on wound inoculation of fruit.

The manuscript is well written however, there are concerns that need to be addressed as follows:

The major concern is that the three NAL isolates are likely NOT to be C. anthrisci but a new species because in the multigene phylogenetic tree there is a significant (100% bootstrap support) clade difference between these isolates and C. anthrisci.  This is contrary to the statement on Line 125 “selected isolates clustered with 100% of bootstrap support with strains of C. anthrisci”. It would have been helpful to check the supplementary files that contain single gene phylogenies however, these could not be downloaded due to an error in the link www.mdpi.com/xxx/s1 . The authors should try Bayesian analysis to see if the clustering changes.

The Type or reference species used in the tree Figure 2 should be indicated with an asterisk or superscript letter.

How can the authors claim from the results in Table 1 that “morphologically, conidia and setae aspects matched with the descriptions of C. anthrisci and differ from C. sambucicola” without any statistical analysis to support this hypothesis?

The pathogenicity bioassay was based on wound inoculated fruit and not surprisingly resulted in disease development.  Wounding the plant tissue prior to inoculating with spore suspension is an inaccurate way to assess a Colletotrichum species pathogenicity and virulence as this method provides a biased measure of pathogenicity. Wounding dismisses the important function of the cuticle waxes and appressoria formed by Colletotrichum isolates before infection. Many Colletotrichum species are unable to infect the cuticle and epidermal cells and thus are not important pathogens. The authors need to reassess pathogenicity using non-wound method of inoculation.  More details are required on the source of the fruit used in the bioassay (farm or supermarket) and age (physiological maturity).  Field grown fruit may already be infected with Colletotrichum spp that remain latent until the fruit ripens; and mature ripe fruit may be more susceptible than immature green fruit.

The Discussion could address the issue of the origin of the pathogen in Chile. Was this species able to infect leaves and twigs of avocado trees or other fruit tree species?

Lines 57, 168, 173. When referring to the species complex (as a Noun) this should be in lower case non italicised eg acutatum species complex NOT (italicised) C. acutatum species complex as this is the Latin description of a specific species.

Line 78. Was the 0.5% sodium hypochlorite the final concentration of the active ingredient?  If so, then insert ‘ai’ after the 0.5%.

Author Response

Dear reviewer,

Thank you for your time and your comments to improve our manuscript. We have accepted all your edits and the changes in the updated version of the text are highlighted in blue.

Please find below our responses to your questions.

  1. The major concern is that the three NAL isolates are likely NOT to be C. anthrisci but a new species because in the multigene phylogenetic tree there is a significant (100% bootstrap support) clade difference between these isolates and C. anthrisci. This is contrary to the statement on Line 125 “selected isolates clustered with 100% of bootstrap support with strains of C. anthrisci”. It would have been helpful to check the supplementary files that contain single gene phylogenies however, these could not be downloaded due to an error in the link www.mdpi.com/xxx/s1 . The authors should try Bayesian analysis to see if the clustering changes.

Response: This issue has been thoroughly discussed among the authors and other researchers that work with Colletotrichum and concluded that the NAL isolates correspond to C. anthrisci based on morphological features and five DNA barcodes (ITS, act, chs-1, his3, and tub2). Since only a few strains of C. anthrisci have been described and characterized, not much is known about its distribution and genetic diversity worldwide. The sequences of the above-mentioned barcodes are identical to C. anthrisci. However, sequences of gapdh indicate that our isolates are closer to C. sambucicola, resulting in an incongruence that is not uncommon in Colletotrichum. Despite of that, to the best of our knowledge, the genetic distance between our isolates and C. anthrisci in the six-locus phylogeny is not enough to consider a new species. A Bayesian inference was also performed which showed the same topology. The BI posterior probabilities were added to the maximum likelihood tree (lines 135-136). Please find attached the supplemental files you could not download.

  1. The Type or reference species used in the tree Figure 2 should be indicated with an asterisk or superscript letter.

Response: Corrected (line 139).

  1. How can the authors claim from the results in Table 1 that “morphologically, conidia and setae aspects matched with the descriptions of C. anthrisci and differ from C. sambucicola” without any statistical analysis to support this hypothesis?

Response: We measured the different morphological features according to the descriptions of both C. anthrisci (Damm et al. 2009) and C. sambucicola (Tibpromma et al. 2017). Data from these studies is not available in a way that is possible to do statistical analysis, but the range and means between the three groups allow to make clear comparisons. For example, conidia dimensions fit the descriptions of C. anthrisci, and not C. sambucicola, that are smaller. Similarly, setae from NAL isolates matched clearly with C. anthrisci, due to unique features of this species such as the constricted base, strongly pointed apex, and length. Line 192 corrected.

  1. The pathogenicity bioassay was based on wound inoculated fruit and not surprisingly resulted in disease development. Wounding the plant tissue prior to inoculating with spore suspension is an inaccurate way to assess a Colletotrichum species pathogenicity and virulence as this method provides a biased measure of pathogenicity. Wounding dismisses the important function of the cuticle waxes and appressoria formed by Colletotrichum isolates before infection. Many Colletotrichum species are unable to infect the cuticle and epidermal cells and thus are not important pathogens. The authors need to reassess pathogenicity using non-wound method of inoculation. More details are required on the source of the fruit used in the bioassay (farm or supermarket) and age (physiological maturity).  Field grown fruit may already be infected with Colletotrichum spp that remain latent until the fruit ripens; and mature ripe fruit may be more susceptible than immature green fruit.

Response: Pathogenicity without wound and on leaves has been performed and the three isolates were pathogenic, with lesions developing slower than in the wounded assay.

  1. The Discussion could address the issue of the origin of the pathogen in Chile. Was this species able to infect leaves and twigs of avocado trees or other fruit tree species?

Response: Yes, the species was able to infect non-wounded leaves. Section 4.6. (line 321) was rewritten to provide more information and details.

  1. Lines 57, 168, 173. When referring to the species complex (as a Noun) this should be in lower case non italicised eg acutatum species complex NOT (italicised) C. acutatum species complex as this is the Latin description of a specific species.

Response: Corrected.

  1. Line 78. Was the 0.5% sodium hypochlorite the final concentration of the active ingredient? If so, then insert ‘ai’ after the 0.5%.

Response: Yes, the 0.5% corresponds to the final concentration of NaOCl.

Reviewer 2 Report

I have reviewed the manuscript entitled “First record of Colletotrichum anthrisci causing anthracnose on 2 avocado fruits in Chile” in a wide perspective. The manuscript present quiet a lot of data and results. I found it very interesting for the characterization of the Colletotrichum anthrisci as principal etiological agents infecting avocado.
The experimental design and its execution are very good and, therefore, the results are relevant. The Morphological characteristics adequately described and results of molecular characterization were supported by phylogenetic relationship among species. I would suggest to add accession numbers of isolates used in current study. A few simple suggestions are as follows:
What was the incidence of disease in surveyed commercial groves?

 How big the diameter of the lesions?

Why wound technique was implemented in pathogenicity? Were there wounds observed on the infected avocado in the field?

What was the relative humidity during pathogenicity test?

A few minor edits, please see the attachment.

Author Response

Dear reviewer,

Thank you for your time and your comments to improve our manuscript. We have accepted all your edits and the changes in the updated version of the text are highlighted in blue.

In the manuscript there were some questions:

  1. Line 70: how much size? (the lesions)

Response: corrected in the manuscript, line 70.

  1. Line 75: how many samples were collected and processed?

Response: Approximately 1,260 fruits (line 235) were sampled in two groves and approximately 10% developed anthracnose symptoms. Out of this subset of symptomatic fruit, 29 fruits developed the unusual sporulation (line 72) associated with C. anthrisci.

  1. Line 255: (typo correction)

Response: The phrase was incomplete and rewritten with details (now line 258-268).

Further, there were more questions in the platform:

  1. What was the incidence of disease in surveyed commercial groves?

Response: Approximately 1,260 fruits (line 240) were sampled in two groves and approximately 10% developed anthracnose symptoms. Out of this subset of symptomatic fruit, 29 fruits developed the unusual sporulation (line 70) associated with C. anthrisci.

  1. How big the diameter of the lesions?

Response: corrected in the manuscript.

  1. Why wound technique was implemented in pathogenicity? Were there wounds observed on the infected avocado in the field?

Response: Fruits were inoculated both with and without wound. Paragraph 4.6. was rewritten to provide more information and details (line 332).

  1. What was the relative humidity during pathogenicity test?

Response: Between 80 and 90%. This information was added to the manuscript.

Reviewer 3 Report

Manuscript 'First record of Colletotrichum anthrisci causing anthracnose on 2 avocado fruits in Chile' is well written and contain valuable information about identification of new fungal pathogen. However, there are some comments:

Material and method section should be before results section.

Lines 254-255. Correct the sentence: Additionally, ripen ‘Hass’ avocado fruits were  inoculated Three representative isolates were analyzed to made it more clear.

Author Response

Dear reviewer,

Thank you for your time and your comments to improve our manuscript. We have accepted your comments and the changes in the updated version of the text are highlighted in blue.

Please find below our responses to your questions.

  1. Material and method section should be before results section.

Response: Yes, however, this is the format of the journal.

  1. Lines 254-255. Correct the sentence: Additionally, ripen ‘Hass’ avocado fruits were inoculated Three representative isolates were analyzed to made it more clear.

Response: Lines 258-268 were added to complete this phrase that was previously incomplete.

Round 2

Reviewer 1 Report

Review 1: The major concern is that the three NAL isolates are likely NOT to be C. anthrisci but a new species because in the multigene phylogenetic tree there is a significant (100% bootstrap support) clade difference between these isolates and C. anthrisci. This is contrary to the statement on Line 125 “selected isolates clustered with 100% of bootstrap support with strains of C. anthrisci”. It would have been helpful to check the supplementary files that contain single gene phylogenies however, these could not be downloaded due to an error in the link www.mdpi.com/xxx/s1 . The authors should try Bayesian analysis to see if the clustering changes.

Response: This issue has been thoroughly discussed among the authors and other researchers that work with Colletotrichum and concluded that the NAL isolates correspond to C. anthrisci based on morphological features and five DNA barcodes (ITS, act, chs-1, his3, and tub2). Since only a few strains of C. anthrisci have been described and characterized, not much is known about its distribution and genetic diversity worldwide. The sequences of the above-mentioned barcodes are identical to C. anthrisci. However, sequences of gapdh indicate that our isolates are closer to C. sambucicola, resulting in an incongruence that is not uncommon in Colletotrichum. Despite of that, to the best of our knowledge, the genetic distance between our isolates and C. anthrisci in the six-locus phylogeny is not enough to consider a new species. A Bayesian inference was also performed which showed the same topology. The BI posterior probabilities were added to the maximum likelihood tree (lines 135-136). Please find attached the supplemental files you could not download.

Review 2

I am still not convinced of the identification of the NAL isolates and suggest that the NAL isolates are more similar to C. sambucicola than C. anthrisci.

-          The morphological descriptors measured do not show the NAL isolates as being more similar to anthrisci than sambucicola (see notes below).

-          There is an incomplete data set of the histone gene sequences (Table S1) for the species in this complex including none for C. sambucicola (Fig S4) which is the important species the NAL isolates are being compared to. The lack of histone data has most likely weakened the phylogeny of the tree and biased the positioning of sambucicola in this tree. I recommend removal of this gene from the study due to lack of data and reconstructing Fig 2 as a five-gene tree – which should still be robust to identify the clade positioning of the NAL isolates.

-          The GAPDH gene is the most informative gene for Colletotrichum and the tree appears to be congruent for the majority of the species with the other trees hence the fact that NAL isolates significantly clade with sambucicola is important.

-          The fact that in Figure 2 NAL isolates are closer to sambucicola indicates that the NAL isolates are indeed sambucicola!

-          The chs-1 gene sequences do not show any support for separating the NAL isolates from the other two species.

-          There appears to be an error with the two actin sequences of C. sambucicola in Fig S1 as the branch root length is disproportionally long and does not have a bootstrap value to signify differences to the NAL and anthrisci isolates.

-          In conclusion, the statement at Line 202 is wrong and needs to be reworded to accurately reflect which gene sequences are aligned with either anthrisci or sambucicola.

Review 1: How can the authors claim from the results in Table 1 that “morphologically, conidia and setae aspects matched with the descriptions of C. anthrisci and differ from C. sambucicola” without any statistical analysis to support this hypothesis?

Response: We measured the different morphological features according to the descriptions of both C. anthrisci (Damm et al. 2009) and C. sambucicola (Tibpromma et al. 2017). Data from these studies is not available in a way that is possible to do statistical analysis, but the range and means between the three groups allow to make clear comparisons. For example, conidia dimensions fit the descriptions of C. anthrisci, and not C. sambucicola, that are smaller. Similarly, setae from NAL isolates matched clearly with C. anthrisci, due to unique features of this species such as the constricted base, strongly pointed apex, and length. Line 192 corrected.

Review 2: -       The metrics for the morphological descriptors measured can not be directly compared to the results in the other publications because different methods were used in separate studies. The only way to provide quantification comparisons would be to grow the two type isolates alongside the Chile isolates on same media, same growing conditions and use the same assessor. Otherwise, there is too much error, and conclusions about morphological differences are inaccurate and no more than a guess. I suggest Table 1 is meaningless and conclusions suggesting NAL isolates are close to one or the other species can not be made. Hence Line 197 “Morphologically, conidia and setae dimensions and features matched with the descriptions of C. anthrisci [19] and differ from C. sambucicola [23].” Is inaccurate and has to be modified to represent the true observations. The authors could argue that there was some similarity to both species.

Review 1: Lines 57, 168, 173. When referring to the species complex (as a Noun) this should be in lower case non italicised eg acutatum species complex NOT (italicised) C. acutatum species complex as this is the Latin description of a specific species.

Response: Corrected.

Review 2: No this was not corrected in Line 57

Author Response

Dear reviewer,
We appreciate your time again in reviewing our manuscript. We have carefully considered your comments and addressed them below. The current version of the manuscript with tracking changes is attached. 

  1. I am still not convinced of the identification of the NAL isolates and suggest that the NAL isolates are more similar to C. sambucicola than C. anthrisci.

Response: Based on new phylogenetic analyses excluding one and two insufficiently informative loci (his3 and chs1), the NAL isolates grouped with 100% of bootstrap support with C. anthrisci strains. More details are provided below.

  1. The morphological descriptors measured do not show the NAL isolates as being more similar to anthrisci than sambucicola (see notes below).

Response: This concern has been responded below.

  1. There is an incomplete data set of the histone gene sequences (Table S1) for the species in this complex including none for C. sambucicola (Fig S4) which is the important species the NAL isolates are being compared to. The lack of histone data has most likely weakened the phylogeny of the tree and biased the positioning of sambucicola in this tree. I recommend removal of this gene from the study due to lack of data and reconstructing Fig 2 as a five-gene tree – which should still be robust to identify the clade positioning of the NAL isolates.

Response: We agree that there is a lack of available his3 sequences for C. sambucicola, and therefore the topology of the tree could be affected by missing data. A five-locus phylogeny (act, chs1, ITS, gapdh, tub2) and a four-locus analysis (act, ITS, gapdh, tub2) were performed and resulting trees showed a 100% clustering with C anthrisci, and the strains of C. sambucicola grouped separately in another clade with 100% support.

  1. The GAPDH gene is the most informative gene for Colletotrichum and the tree appears to be congruent for the majority of the species with the other trees hence the fact that NAL isolates significantly clade with sambucicola is important.

Response: We disagree with the first statement. There is no single most informative gene in Colletotrichum (Jayawardena et al. 2016, Vieira et al. 2017; Vieira et al. 2020). For the dematium complex, the combination of gapdh, his3 and tub2 provide the best resolution. Plus, even though act has been considered not as informative, the act sequences of NAL isolates are identical to C. anthrisci (100%), and significantly different than C. sambucicola (86% identity).

  1. The fact that in Figure 2 NAL isolates are closer to sambucicola indicates that the NAL isolates are indeed sambucicola!

Response: Not necessarily. The topology was different in the single-locus phylogenies. When removing non-informative loci, such as his3 and chs1, NAL isolates clustered with C. anthrisci only, and not with C. sambucicola.

  1. The chs-1 gene sequences do not show any support for separating the NAL isolates from the other two species.

Response: Yes, when comparing the chs-1 sequences between NAL isolates, C. anthrisci and C. sambucicola no differences were found, which was evident in the single-gene phylogeny. For this reason, chs-1 was included in the new five-gene phylogeny and excluded in an alternative four-gene phylogeny. Resulting trees showed that NAL isolates clustered with 100% of support with strains of C. anthrisci, and not with C. sambucicola.

  1. There appears to be an error with the two actin sequences of C. sambucicola in Fig S1 as the branch root length is disproportionally long and does not have a bootstrap value to signify differences to the NAL and anthrisci isolates.

Response: The actin sequences were double-checked, and they were correct in the original analysis. An explanation for the long branch of C. sambucicola strains is that there is a 39 bp difference when compared to the sequences of C. anthrisci strains and NAL isolates , which is higher than the nucleotide differences with ITS and tub2.

  1. In conclusion, the statement at Line 202 is wrong and needs to be reworded to accurately reflect which gene sequences are aligned with either anthrisci or sambucicola.

Response: Line 202 has been updated with the new five-locus tree.

  1. The metrics for the morphological descriptors measured can not be directly compared to the results in the other publications because different methods were used in separate studies. The only way to provide quantification comparisons would be to grow the two type isolates alongside the Chile isolates on same media, same growing conditions and use the same assessor. Otherwise, there is too much error, and conclusions about morphological differences are inaccurate and no more than a guess. I suggest Table 1 is meaningless and conclusions suggesting NAL isolates are close to one or the other species can not be made. Hence Line 197 “Morphologically, conidia and setae dimensions and features matched with the descriptions of C. anthrisci [19] and differ from C. sambucicola [23].” Is inaccurate and has to be modified to represent the true observations. The authors could argue that there was some similarity to both species.

Response: We understand that morphological features cannot be compared when the structures of NAL isolates and the strains of C. anthrisci and C. sambucicola have been measured under different conditions. However, critical features that have been described to be unique to C. anthrisci (and different than C. sambucicola) have been confirmed with NAL isolates. These features are the following: (1) conidia length above 25 µm, as opposed to 17.7 µm in average found in C. sambucicola; (2) conidia with strongly acute apex, as opposed to the slight acute to rounded apex; (3) setae length that ranged from 112.5 to 480 µm, compared to the C. sambucicola that are notably shorter (110-145 µm); (4) appressoria present, with bullet-shape to navicular, when in C. sambucicola, they were not observed nor described. Therefore, in terms of the narrative of the manuscript, we could argue that even though there were similarities to both species, there is a tendency to resemble more the characteristics of C. anthrisci. Lines 196-198 have been rephrased.

  1. No this was not corrected in Line 57.

Response: It has been corrected now.

Round 3

Reviewer 1 Report

The revised phylogenetics are sound and provides a convincing argument that the species is anthrisci. The authors have made changes in the text to consider previous concerns.